# Deoxynivalenol Induces Inflammation in IPEC-J2 Cells by Activating P38 Mapk And Erk1/2

**DOI:** 10.3390/toxins12030180

**Published:** 2020-03-13

**Authors:** Hua Zhang, Xiwen Deng, Chuang Zhou, Wenda Wu, Haibin Zhang

**Affiliations:** 1College of Veterinary Medicine, Nanjing Agricultural University, Nanjing 210095, China; 2016207033@njau.edu.cn (H.Z.); 9171310404@njau.edu.cn (X.D.); 2Jiangsu Vocational College of Agriculture and Forestry, Jurong 212400, China; zhouchuang@jsafc.edu.cn; 3Department of Chemistry, Faculty of Science, University of Hradec Kralove, 500 03 Hradec Kralove, Czech Republic

**Keywords:** deoxynivalenol, IPEC-J2 cells, RNA-seq, inflammation, MAPKs

## Abstract

Fusarium-derived mycotoxin deoxynivalenol (DON) usually induces diarrhea, vomiting and gastrointestinal inflammation. We studied the cytotoxic effect of DON on porcine small intestinal epithelium using the intestinal porcine epithelial cell line IPEC-J2. We screened out differentially expressed genes (DEGs) using RNA-seq and identified 320 upregulated genes and 160 downregulated genes. The enrichment pathways of these DEGs focused on immune-related pathways. DON induced proinflammatory gene expression, including cytokines, chemokines and other inflammation-related genes. DON increased IL1A, IL6 and TNF-α release and DON activated the phosphorylation of extracellular signal-regulated kinase-1 and-2 (ERK1/2), JUN N-terminal kinase (JNK) and p38 MAPK. A p38 inhibitor attenuated DON-induced IL6, TNF-α, CXCL2, CXCL8, IL12A, IL1A, CCL20, CCL4 and IL15 production, while an ERK1/2 inhibitor had only a small inhibitory effect on IL15 and IL6. An inhibitor of p38 MAPK decreased the release of IL1A, IL6 and TNF-α and an inhibitor of ERK1/2 partly attenuated protein levels of IL6. These data demonstrate that DON induces proinflammatory factor production in IPEC-J2 cells by activating p38 and ERK1/2.

## 1. Introduction

Deoxynivalenol (DON; vomitoxin) is a type B trichothecene mycotoxin produced by strains of *Fusarium graminearum* and *F. culmorum* [1]. DON mainly contaminates cereal, especially barley, oats, wheat, corn and their subsequent products. In addition, DON accumulation is a potential sign for the occurrence of other mycotoxins [2]. Due to its adverse effects on animals, DON is known as one of the most significant mycotoxins in animal production.

Consumption of DON-contaminated foods and feeds has been associated with a spectrum of adverse effects and the immunotoxic effects of DON are of increasing concern for farm animals, as well as for humans [2,3]. According to the dose, timing of exposure, time and functional immune assay being used, DON may exert immunosuppressive or immunostimulatory effects [4]. Our preliminary experiments indicate that exposure to DON induces the overexpression of cytokines and chemokines, leading to immune stress, which caused immune function damage [5,6].

The intestinal epithelium forms an important physical barrier against external matter and it is highly sensitive to mycotoxins and important for maintaining health [7]. Consuming DON-contaminated food is related to gastroenteritis flare-ups and DON exposure leads to intestinal lesions in vivo (animals studies), ex vivo (intestinal explants) and in vitro (cell line) [8,9,10]. Numerous studies have concluded that DON upregulates the expression of cytokines, chemokines and inflammatory genes [11,12,13]. However, the mechanism underlying DON-induced inflammation in intestinal epithelial cells (IECs) remains unclear.

MAPKs, including p38 MAPK, extracellular signal-regulated kinase-1 and-2 (ERK1/2) and JUN N-terminal kinase (JNK), modulate many cellular processes associated with cell proliferation, differentiation, survival and death [14]. MAPK signaling has basic functions in immunoregulation and immunopathology, including inflammatory responses and enteritis. Recent research has suggested that DON and other trichothecenes induce the activation of MAPKs in IPEC-J2 cells [15,16,17], which contributes to autophagy, oxidative stress, epithelial tight junction disruption and intestinal barrier dysfunction. However, few correlative studies have investigated the interaction of MAPK signaling with DON-induced inflammation in the intestinal epithelium.

Therefore, the aims of the present study were to use the IPEC-J2 cell line, an in vitro model of porcine small IECs, to investigate the capacity of DON to induce inflammation and relate the immunomodulatory effects of DON to MAPK activation.

## 2. Results

### 2.1. DON Decreases the Viability and Induces Inflammation in IPEC-J2 Cells

IPEC-J2 cells were treated for different time periods (2, 6, 12 and 24 h) and with different concentrations of DON (0.25, 0.5, 1, 2 and 4 μg/mL). As presented in Figure 1a, DON (≥0.5 μg/mL) significantly reduced IPEC-J2 cell viability in a time- and concentration-dependent manner.

DON at concentrations of 1.0 and 2.0 μg/mL markedly enhanced the gene expression levels of IL6, IL1A and TNF-α at 2 h compared to the control group (Figure 1b,d,f). After treatment with DON at concentrations of 1.0 and 2.0 μg/mL, the expression of IL1A and IL6 was significantly increased at 6 h (Figure 1b,f) and the expression of IL6 was significantly increased at 12 h (Figure 1b). Moreover, IL6, IL1A and TNF-α protein release into the incubation medium was elevated after treatment with DON at concentrations of 1.0 and 2.0 μg/mL (Figure 1c,e,g). To investigate the immunomodulatory effects of DON, IPEC-J2 cells were exposed to 2 μg/mL DON for 2 h in subsequent experiments.

### 2.2. Identification and Functional Enrichment Analysis of Differentially Expressed Genes (DEGs)

Based on the RNA-seq data, we obtained 480 differentially expressed genes (DEGs) with 320 upregulated genes and 160 downregulated genes (Appendix A). In Figure 2, the heatmap and volcano plot show that these genes were clearly separated (Figure 2a,b). According to the Gene Ontology (GO) terms (Figure 2c), 71 genes were enriched in the immune system process. Kyoto Encyclopedia of Genes and Genomes (KEGG) pathway analysis revealed that the upregulated DEGs were mainly enriched in the following immune-related pathways: TNF signaling pathway, cytokine-cytokine receptor interaction, MAPK signaling pathway, NF-kappa B signaling pathway, Jak-STAT signaling pathway, Toll-like receptor signaling pathway and NOD-like receptor signaling pathway (Figure 2d). Table 1 shows several enriched pathway terms and 15 DEGs were enriched in the MAPK signaling pathway. These results suggest that DON-induced inflammation may associate with the MAPK signaling pathway.

### 2.3. Integration of Protein-Protein Interaction (PPI) Network Analysis

To further investigate regulatory pathways of DON, a protein-protein interaction (PPI) network was formulated based on the data in the Search Tool for the Retrieval of Interacting Genes/Proteins (STRING) database with a total of 371 nodes and 729 relationship pairs (Figure 3a). The top 10 hub genes were TNF, IL6, JUN, MYC, CXCL8, FOS, EGR1, CSF2, EDN1 and ATF3 and they were key node proteins in the PPI network. To better analyze the interaction of the proteins, we detected two modules using the Cytoscape plugin Molecular Complex Detection (MCODE) with a score >5 and the top module is shown in Figure 3b. Pathway enrichment analysis of the top module showed that it was mainly related to the MAPK signaling pathway, cytokine-cytokine receptor interaction and TNF signaling pathway.

### 2.4. Validation of the Expression Profile Analysis by RT-qPCR

Ten genes were selected from the significant DEGs for RT-qPCR analysis to validate their expression levels. The transcriptional levels according to the sequencing and RT-qPCR data were consistent (Figure 2e), thus confirming that the sequencing information was reliable.

### 2.5. DON Promotes the Expression of Inflammatory Factors and Induces Inflammation in IPEC-J2 Cells Through p38 and ERK1/2

We hypothesized that there may be a link between the activation of the MAPK pathway and DON-induced inflammation. We measured the phosphorylated protein levels of p38, ERK1/2 and JNK. DON effectively increased the phosphorylation of p38, ERK1/2 and JNK (Figure 4).

To gain insight into the mechanism of MAPKs in DON-induced inflammatory factor upregulation, IPEC-J2 cells were pretreated with inhibitors, including U0126 (ERK 1/2 inhibitor, 10 mM), SP600125 (JNK inhibitor, 20 mM) and SB203580 (p38 inhibitor, 10 mM), before DON treatment. As shown in Figure 5, the inhibition of p38 significantly attenuated DON-induced IL6, TNF-α, CXCL2, CXCL8, IL12A, IL1A, CCL20, CCL4 and IL15 production, whereas the inhibition of JNK had no effect. In addition, the inhibition of ERK 1/2 attenuated DON-induced IL15 and IL6 production. In contrast, CCL4, CCL20 and CXCL2 production increased after treatment with the ERK 1/2 and JNK inhibitors. DON treatment did not significantly affect CCL2 production. In addition, the inhibition of p38 significantly attenuated IL6, IL1A and TNF-α protein release and the inhibition of ERK 1/2 partly attenuated DON-induced IL6 protein release (Figure 6). These results suggest that both p38 and ERK 1/2 contribute to DON-induced inflammation.

## 3. Discussion

The mycotoxin DON is a frequent contaminant of cereals and co-products. The intestine, which serves as the first barrier against food contaminants, shows high sensitivity to DON and related mycotoxins [7,13,18]. After pigs are exposed to DON, most absorption occurs in jejunal epithelial cells. DON mainly causes oxidative stress, disrupts epithelial tight junctions and induces intestinal barrier dysfunction [15,17]. However, the mechanism underlying DON-induced inflammation in IECs is not completely clear. To gain insight into the genes and pathways related to DON in IPEC-J2 cells, we conducted RNA-seq analysis to identify the top inflammatory factors and molecular pathways following DON treatment.

DON robustly upregulates proinflammatory gene expression [4]. DON increased the expression of genes and proteins associated with inflammation, such as TNF-α and IL6 in IPEC-J2 cells, which was consistent with a previous study [19]. TNF-α and interleukins are classic proinflammatory factors that are quickly secreted and cause inflammation when the body is exposed to exogenous stimulation [20]. Overabundant production of TNF-α causes excess secretion of other inflammatory factors, such as IL1β, IL2 and IL8, thereby inducing intestinal mucosal injury [20,21,22]. Accordingly, inflammatory factors play roles in intestinal immunity. Our data showed that DON significantly upregulated the levels of proinflammatory factors in a concentration-dependent manner in IPEC-J2 cells, indicating that DON enhances the production of inflammatory mediators.

Apart from proinflammatory cytokine upregulation, DON upregulates the transcription levels of several chemokines, including CXCL2, CCL2 and CCL20 [6,23,24]. In our study, DON upregulated the chemokines CXCL2, CXCL8, CCL4 and CCL20. The chemokine CXCL2 is a cytokine secreted by IPEC-J2 cells and a chemotactic for polymorphonuclear leukocytes [25]. CXCL8 is a proinflammatory chemokine that acts as a strong chemoattractant but can create tissue injury with long-term exposure [26]. CCL4 serves as a chemoattractant for monocytes, natural killer cells and a variety of other immune cells [27] and CCL20 is strongly chemotactic for lymphocytes [28]. DON induces the release of CXCL8 in several intestinal epithelial cell lines [29,30]. These previous results are in agreement with our study. Thereby, the inflammation effects of DON may, in part, be influenced by the leukocyte chemotaxis induced by chemokine dysfunction.

The KEGG pathway enrichment analysis showed that the significant DEGs were enriched in immunological pathways and that the MAPK signaling pathway was one of the main signaling pathway enriched in 15 DEGs. Pathway enrichment analysis of the top module showed that it was mainly associated with MAPKs. MAPKs are a type of protein kinase that is pivotal for the development of inflammation [31]. MAPK pathways are activated by kinases, cytoskeletal proteins, transcription factors and other enzymes [32]. The first step to their activation consists of relieving their autoinhibition by a smaller ligand (such as Ras for c-Ra and GADD45 for MEKK4) [33]. DUSPs negatively regulate some MAPKs. DUSP5 and DUSP6 inactivate ERK1/2 and DUSP1 interacts with p38-α, ERK2 and JNK1 [34,35]. MAP3K8, MAP3K5 and MAP3K14 are important MAP3 kinases. The transcription factors JUN, MYC and FOS regulate the expression of inflammation- and immune-related genes [4]. In our study, the upregulation of GADD45B, GADD45G, RASA1, MAP3K8, MAP3K14, MAP3K5, IL1A, MYC, FOS, TNF and JUN, which are related to the MAPK pathway, contributed to MAPK activation and the expression of inflammatory factors. According to the RNA-seq analysis, DON may induce inflammation via the MAPK pathway.

MAPK contributes to DON-induced transactivation and the mRNA stabilization of inflammatory factors [36,37]. To determine whether DON induces porcine intestinal epithelium cell inflammation via the MAPK pathway, MAPK inhibition assays were performed. It has been reported that the MAPK pathway is one of the main pathways for DON to induce inflammation [11,22]. The results in the present study showed that DON induced activation of MAPKs. The p38 inhibitor attenuated DON-induced gene expression levels of IL6, TNF-α, CXCL2, CXCL8, IL12A, IL1A, CCL20, CCL4 and IL15 as well as protein expression levels of IL1A, IL6 and TNF-α. The ERK1/2 inhibitor had only a small inhibitory effect on IL1A and IL6 gene expression levels as well as IL6 protein levels, while the JNK inhibitor had no effect. We demonstrated that DON induced the expression of proinflammatory cytokines and chemokines via the p38 MAPK and ERK1/2 signaling pathways. CXCL8 secretion were upregulated in various human intestinal epithelial cell lines exposed to DON [29,30,38]. In response to DON, dose-dependent increases in IL-8 secretion were observed in Caco-2 cells and this was linked to the ribotoxic-associated activation of PKR, NF-kB and p38 [29,38]. DON elevates CXCL8 generation via ERK1/2 but not p38 in human embryonic epithelial intestine 407 (Int407) cells [30]. This discrepancy may be due to the maturation status of the cells: differentiated mature Caco-2 cells and IPEC-J2 vs. undifferentiated Int407 cells.

In conclusion, the results of the present study indicate that DON induces inflammation in IPEC-J2 cells. This discovery provides a theoretical basis for further exploring the molecular mechanisms of IEC inflammation induced by DON.

## 4. Materials and Methods

### 4.1. Reagents

DON was obtained from Sigma-Aldrich (St. Louis, MO, USA). Cell culture medium and supplements were purchased from Life Technologies (Grand Island, NY, USA). Anti-phospho-p38 (4511), anti-p38 (8690), anti-phospho-JNK (4668), anti-JNK (9252), anti-phospho-ERK (4370), anti-ERK (4695) and anti-β-actin (4970) antibodies were purchased from Cell Signaling Technology (Beverly, MA, USA). SB203580 was obtained from Promega (Madison, WI, USA). U0126 and SP600125 were acquired from Cayman Chemicals (Ann Arbor, MI, USA).

### 4.2. Cell Culture and Treatment

The IPEC-J2 cell line was a gift from Professor Qian Yang, Nanjing Agricultural University, Nanjing, China. Cells were grown in DMEM/F12 medium supplemented with antibiotics and 10% fetal bovine serum. Cells were maintained in the exponential growth phase by passages at intervals of 2–3 days. Compounds were prepared as stock solutions and diluted with the cell culture medium before use. The working concentrations were as follows: DON (0.25, 0.5, 1, 2 and 4 μg/mL), U0126 (10 μM), SP600125 (20 μM) and SB203580 (10 μM). The final concentration of dimethyl sulfoxide (DMSO) was less than 0.1%, which exerted no effect on cell viability. Cells were treated with or without DON and the indicated test compounds for various times according to the experimental protocol.

### 4.3. Cell Viability Assay

Cell viability was measured using the MTT (Sigma, M5655) method according to the manufacturer’s instructions after DON treatments for 2, 6, 12 and 24 h. The optical density of the control group was considered to be 100% viable.

### 4.4. Quantitative Real-Time PCR (qRT-PCR) Assay

Total RNA was isolated using TRIzol reagent (Takara, Dalian, China). cDNA was obtained by reverse transcription using a cDNA transcription kit (Takara, Dalian, China). Real-time PCR was performed in 96-well optical plates on an ABI StepOne Plus Real-time PCR system using SYBR Premix Ex Taq™ (Takara, Dalian, China). The primers used for RT-PCR are shown in Table 2. Analysis of the relative gene expression level was achieved using the 2^-ΔΔCT^ method and gene expression levels were normalized to GAPDH.

### 4.5. Cytokine Detection by ELISA

IPEC-J2 cell supernatants were collected after treatment with 2.0 mg/mL DON for 12 h. Porcine IL6, IL1A and TNF-α ELISAs (MEIMIAN, Jiangsu, China) were performed according to the manufacturer’s instructions. Samples were analyzed in duplicate.

### 4.6. RNA-seq Analysis

After 2 h of exposure to DON, IPEC-J2 cells were collected. Total RNA was extracted using the miRNeasy Mini Kit (Qiagen, Hilden, Germany) following the manufacturer’s instructions. cDNA library construction and sequencing with an Illumina HiSeq 2000 sequencer were performed at Shanghai Biotechnology Corporation (Shanghai, China). The resulting RNA-seq reads were mapped onto the reference genome of Sscrofa11.1. The generated RNA-seq data were deposited in the National Center for Biotechnology Information (NCBI) Sequence Read Archive (SRA) repository with accession number PRJNA578240. The expression of transcripts was quantified as fragments per kilobase of exon model per million mapped reads (FPKM). Genes with differential expression levels were identified using edgeR [39]. Differential expression P-values were false discovery rate (FDR)-adjusted using the q-value Bioconductor package. Genes with a q-value ≤0.05 and |fold change| ≥2 were defined as differentially expressed. We analyzed the enrichment of the DEGs using GO functional enrichment analysis and KEGG pathway analysis. ClusterProfiler is a R package applied to perform GO function and KEGG pathway enrichment analyses on DEGs. The terms were considered to be significantly enriched if q-value ≤ 0.05.

### 4.7. PPI Network Analysis

The STRING online tool (https://string-db.org/cgi/input.pl) was used to construct a PPI network of the DEGs with a confidence score >0.4 defined as significant [40]. We then imported the interaction data into Cytoscape (version 3.6.0, http://chianti.ucsd.edu/cytoscape-3.6.0/) to map the PPI network [41]. The MCODE plugin for Cytoscape was used to analyze the interaction relationships of the DEGs with their encoded proteins and to screen the hub genes.

### 4.8. Western Blot Analyses

After 2 h of exposure to DON, IPEC-J2 cells were collected and lysed in cell lysis buffer (Beyotime, Haimen, China). Protein concentrations were determined using a BCA protein assay kit (Beyotime, China). Proteins were separated by electrophoresis and transferred to PVDF membranes. Anti-phospho-p38 (1:1000), anti-p38 (1:1000), anti-phospho-JNK (1:1000), anti-JNK (1:1000), anti-phospho-ERK (1:2000), anti-ERK (1:1000) and anti-β-actin antibodies were used as primary antibodies. Proteins bound by the primary antibodies were visualized with an appropriate secondary antibody (1:5000) and then detected by an ECL Chemiluminescence kit (Vazyme, E411-05). Protein bands were quantified using NIH ImageJ software (available in the public domain) and detected using a Bio-Rad imaging system (Bio-Rad, Hercules, CA, USA).

### 4.9. Statistical Analysis

All data were statistically analyzed using SigmaPlot 11 for Windows (Jandel Scientific; San Rafael, CA, USA). Data of cell viability were analyzed by a two-way ANOVA using the Holm–Sidak method. Other test were assessed by one-way ANOVA with Holm-Sidak tests. Data were considered to be statistically significant difference if *p* < 0.05.

## Figures and Tables

**Figure 1 toxins-12-00180-f001:**
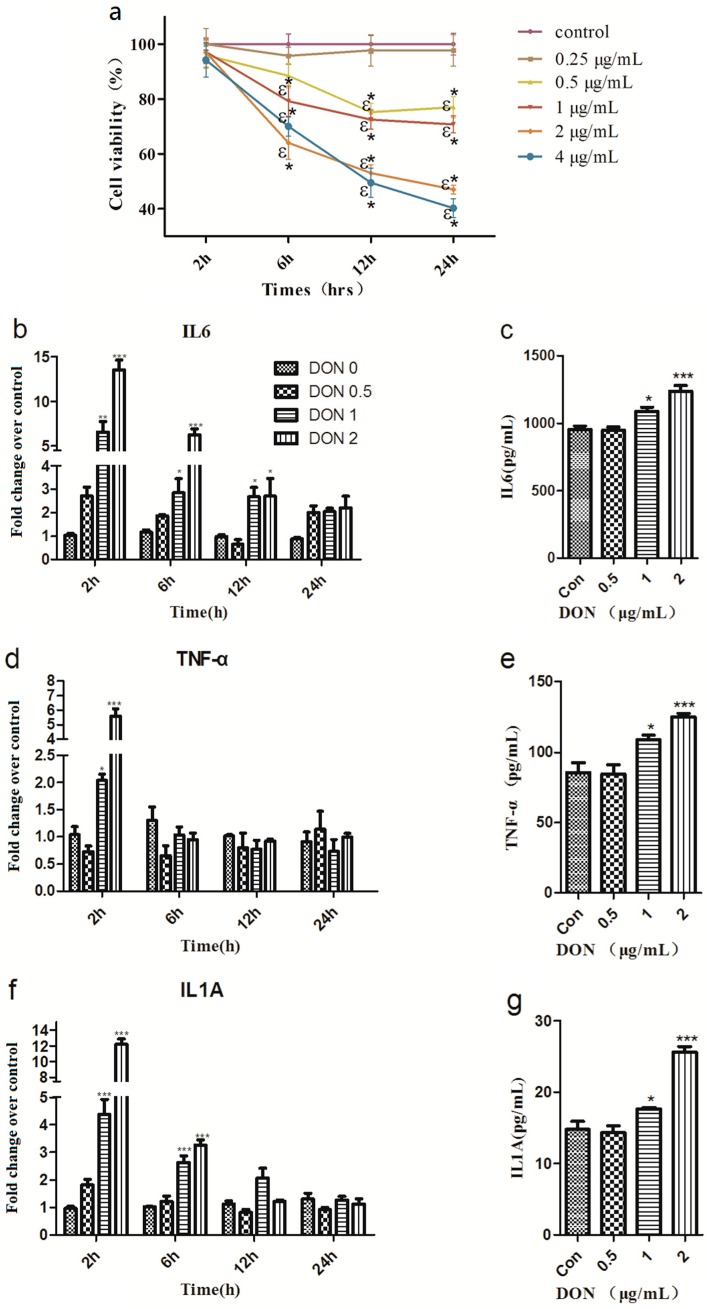
Deoxynivalenol (DON) decreases the viability and induces inflammation in IPEC-J2 cells (**a**) Cell viability in IPEC-J2 cells with or without DON. Two-way ANOVA using Holm-Sidak method was used to assess significant differences in cell viability compared with of the control. Symbols: * indicates difference in cell viability relative to the control at specific time point (*p* < 0.05) and ε indicates difference in cell viability relative to the 2h exposure time at specific dose (*p* < 0.05). Effects of DON on IL6 (**b**), TNF-α (**d**) and IL1A (**f**) gene expression. and IL6 (**c**), TNF-α (**e**) and IL1A (**g**) cytokine release in IPEC-J2 cells. Samples were collected after 2, 6, 12 and 24 h (mRNA) or 12 h (protein release). One-way ANOVA with a Holm-Sidak test was used to assess significant differences in the mRNA and protein release of IL6, TNF-α and IL1A compared with of the control. The data are expressed as the mean ± SEM. * *p* < 0.05, ** *p* < 0.01 and *** *p* < 0.001 versus control.

**Figure 2 toxins-12-00180-f002:**
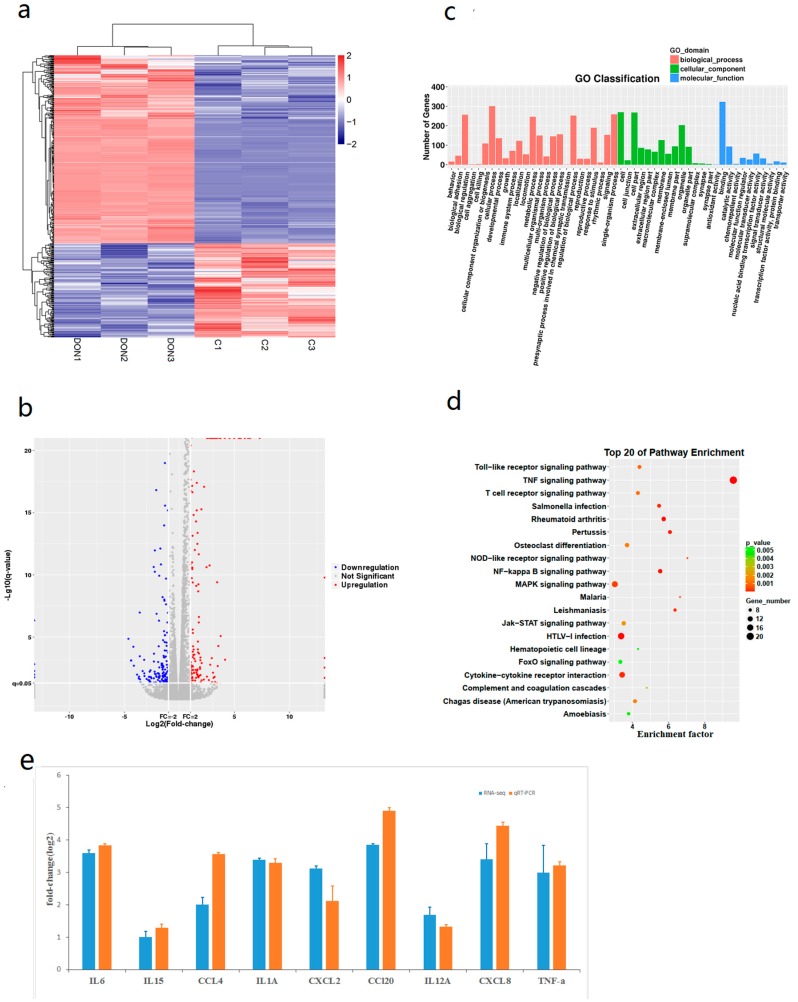
(**a**) Cluster heatmap. A change in color from blue to red indicates that the expression level of the gene was relatively high. (**b**) Volcano plot of the DEGs. Blue indicates downregulated genes and red indicates upregulated genes. (**c**) Gene ontology (GO) analysis classified the DEGs into 3 groups: molecular function, biological process and cellular component. (**d**) Bubbles of Kyoto Encyclopedia of Genes and Genomes (KEGG) pathways of the DEGs. The coloring indicates higher enrichment in red and lower enrichment in green. The point size indicates the number of DEGs enriched in a certain pathway. Lower q-values indicate more significant enrichment. (**e**) Validation of DEG data by real-time quantitative PCR (RT-qPCR). The x-axis represents the mRNAs and the y-axis is the fold change between the RT-qPCR and sequencing values.

**Figure 3 toxins-12-00180-f003:**
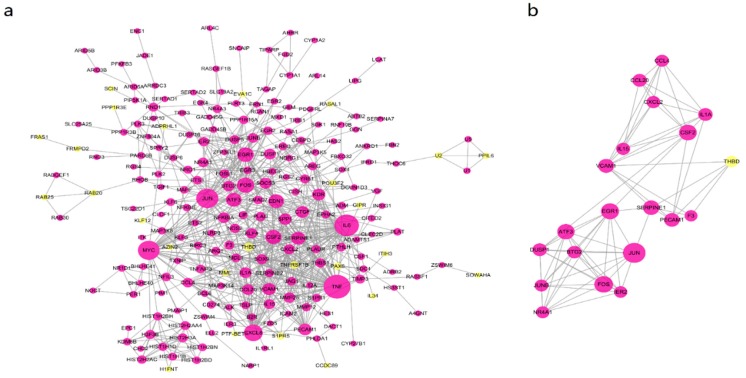
Protein-protein interaction (PPI) network of the DEGs (**a**) and the most significant modules (**b**). Purple nodes represent upregulated genes and yellow nodes represent downregulated genes.

**Figure 4 toxins-12-00180-f004:**
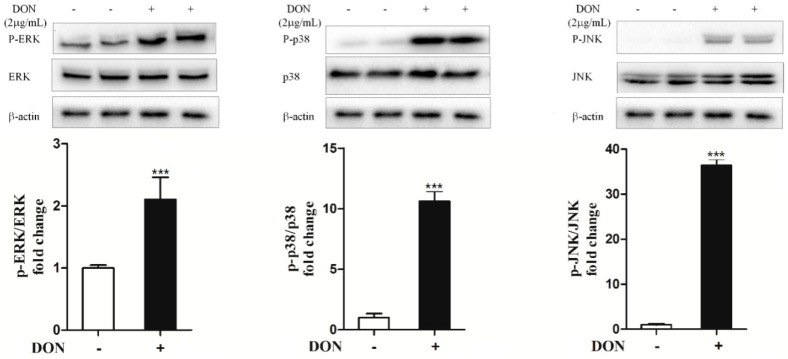
DON induces MAPK activation in IPEC-J2 cells. The levels of p-ERK, p-p38 and p-JNK were detected by western blotting. Data analyzed as described in Figure 1b legend. The quantitative data are presented as the mean ± SEM. * *p* < 0.05, ** *p* < 0.01 and *** *p* < 0.001 versus control.

**Figure 5 toxins-12-00180-f005:**
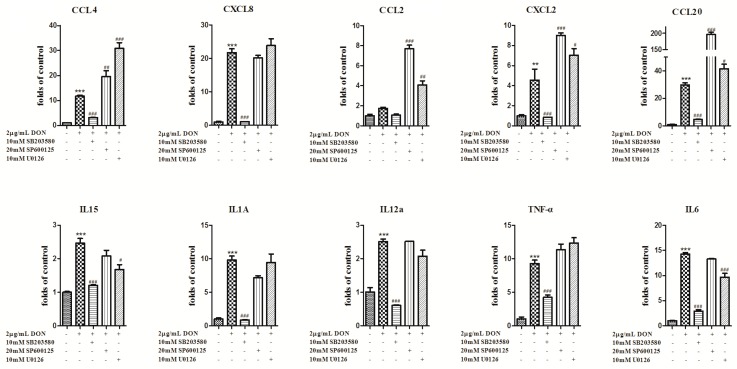
DON promotes the expression of inflammatory factors through p38 and ERK1/2. Data analyzed as described in Figure 1b legend. The data are expressed as the mean ± SEM. * *p* < 0.05, ** *p* < 0.01 and *** *p* < 0.001 versus control. ^#^
*p* < 0.05, ^##^
*p* < 0.01 and ^###^
*p* < 0.001 versus control-DON.

**Figure 6 toxins-12-00180-f006:**
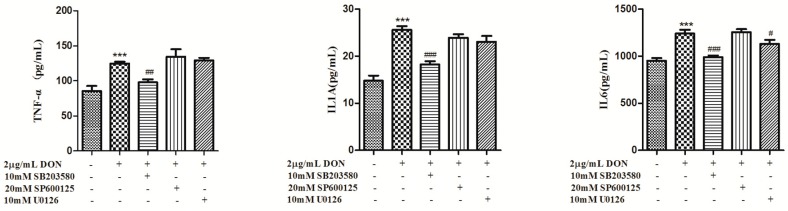
DON induces inflammation in IPEC-J2 cells through p38 and ERK1/2. Data analyzed as described in Figure 1b legend. The data are expressed as the mean ± SEM. * *p* < 0.05, ** *p* < 0.01 and *** *p* < 0.001 versus control. ^#^
*p* < 0.05, ^##^
*p* < 0.01 and ^###^
*p* < 0.001 versus control-DON.

**Table 1 toxins-12-00180-t001:** Pathway enrichment analysis of the differentially expressed genes (DEGs).

Pathways	Number	Gene upregulation	Gene downregulation	Q-value
TNF signaling pathway	21	TNFAIP3, MAP3K8, CCL20, CSF2, LIF, EDN1, CXCL2, IL15, NFKBIA, FOS, MAP3K14, CSF1, TNF, IL6, VCAM1, JUN, MAP3K5, PTGS2, SOCS3, BIRC3, JUNB	-	2.03 × 10^−14^
HTLV-I infection	17	FZD5, EGR1, CSF2, ATF3, MYC, IL15 NFKBIA, FOS, MAP3K14, TNF, IL6, VCAM1, FOSL1, JUN, EGR2, ETS1, ETS2	-	3.47 × 10^−05^
MAPK signaling pathway	15	RASA1, GADD45G, DUSP1, MAP3K8, DUSP5, IL1A, GADD45B, MYC, FOS, MAP3K14, TNF, JUN, DUSP10, MAP3K5, DUSP6	-	0.000375
Cytokine-cytokine receptor interaction	13	CCL4, IL6, IL1A, CCL20, CSF2, KDR, TSLP, IL12A, LIF, IL15, CSF1, TNF, CXCL8	-	0.000191
NF-kappa B signaling pathway	10	VCAM1, TNF, PTGS2, CXCL8, NFKBIA, BIRC3, PLAU, CCL4, TNFAIP3, MAPK3K14	-	6.84 × 10^−05^
Jak-STAT signaling pathway	10	CSF2, TSLP, IL12A, LIF, MYC, MCL1, IL15, PIM1, IL6, SOCS3	-	0.001979
Rheumatoid arthritis	10	IL1A, CCL20, CSF2, IL15, FOS, CSF1, TNF, IL6, CXCL8, JUN	-	5.21 × 10^−05^
Toll-like receptor signaling pathway	9	CCL4, IL12A, NFKBIA, FOS, TNF, IL6, CXCL8, JUN, SPP1	-	0.000926

**Table 2 toxins-12-00180-t002:** Primer sequences of RT-PCR target genes.

Gene	Primer sequence (5′-3′)	
	Sense	Antisense
GAPDH	CGTCAAGCTCATTTCCTGGT	TGGGATGGAAACTGGAAGTC
IL6	AGCAAGGAGGTACTGGCAGA	CAGGGTCTGGATCAGTGCTT
TNF-α	AACCTCCTCTCTGCCATCAA	TAGACCTGCCCAGATTCAGC
CXCL2	CACAGACCCTCCGAGCTAAG	TGACTTCCGTTTGGTCACAG
CXCL8	GCCTCATTCCTGTGCTGGTCAG	AACAACGTGCATGGGACACTGG
IL12A	AAGCCCTCCCTGGAAGAACTGG	TCACCGCACGAATTCTGAAGGC
IL1A	CGAACCCGTGTTGCTGAAGGAG	TGGATGGGCGGCTGATTTGAAG
CCL20	GATGTCGGTGCTGCTGCTCTAC	ATTGGCGAGCTGCTGTGTGAAG
CCL2	CACCAGCAGCAAGTGTCCTA	GGGCAAGTTAGAAGGAAATGAA
CCL4	TGGTCCTGGTCGCTGCCTTC	TTCCGCACGGTGTATGTGAAGC
IL15	TGCATCCAGTGCTACTTGTGTT	GACCTGCACTGATACAGCCC

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
