# Peer review of "Deoxynivalenol Induces Inflammation in IPEC-J2 Cells by Activating P38 Mapk And Erk1/2"

_toxins, 2020, doi:10.3390/toxins12030180_

Round 1
Reviewer 1 Report
The presented study describes the results of an RNA-seq experiment of DON-treated IPEC-J2 cells versus control cells and some downstream confirmation experiments. The study appears to be well executed, simple and straightforward. I have mostly minor comments on some analyses and data presentation.
Major comments:
1) The GO and KEGG enrichment analyses are not described in sufficient detail. What methods were used to perform these analyses? What test sets were used? It appears from the results that the authors analyzed up-regulated and down-regulated DEG sets differently. I am not sure that is appropriate. In a pathway there are typically some genes that inhibit others, meaning that, for example, down-regulation of an inhibitor leads to up-regulation of its targets. In this case, down-regulation of the inhibitor would be important for pathway function. To properly assess the experimental effect, up- and down-regulated genes need to be analyzed together. The authors should therefore repeat the analysis this way. I’m not saying the analysis as done is wrong, but also doing it the other way could add important details.
Minor comments:
2) Figure 1A has a lot of problems: Why is Figure 1A separated from the remaining panels? Why does the viability scale in Figure 1A go to 150%? There cannot be a viability >100%, so the axis should stop at 100%. The x-axis is linear, but the data is exponential. Therefore, the axis should be changed to exponential. I think it would make more intuitive sense to plot time on the x-axis and encode DON concentration in different symbols/colors, even though there are more concentrations than time points.
3) Statistical significance in the cell viability assay should not be calculated as separate tests per time point and concentration. Instead, these should be factors in a single, multivariate testing procedure (two-way ANCOVA).
4) Line 82-83: “These results suggest that DON-induced inflammation was associated with the MAPK signaling pathway.” No, they do not. At least not more than several other enriched GO terms. TNF signaling is much more strongly enriched than MAPK signaling. At this point, without the results presented in Figure 5 below, there is no reason to say that MAPK signaling is more enriched than others. Besides, MAPK signaling is one of several ways of how TNF signaling can be resolved in a cellular response, others being NF-kappa B signaling and Jak-STAT signaling, which are also enriched at similar levels as MAPK signaling.
Similarly, the sentence in lines 167-169 needs to be rephrased as it wrongly emphasizes MAPK signaling as the major outcome from the KEGG analysis, which is incorrect.
5) The legend of Figure 2c says this shows a “GO terms of biological function analysis.” This needs to be elaborated. What contrast is being used? Does it include all expressed genes or only DEGs? The term “biological function” reminds me of “molecular function” and “biological processes”, both of which are distinct GO subcategories, but all three of the subcategories are apparently tested. This should be phrased more cautiously.
6) Error bars should be provided for Figure 2e. Both qPCR and RNA-seq have been done in replicates, so error estimates for both techniques exist and should be provided.
7) Lines 102-101: The data underlying this sentence should be provided.
8) Line 149: Source 7 references a review paper, not a study. If the authors intend to refer to a certain study, this original study should be cited instead of the review paper. Citing the review paper is only appropriate if it summarizes the results from several studies. In that case, the sentence needs to be rephrased to reflect this meaning.
9) Line 192: “The discrepancy is possibly because of the cells origin.” Are the authors trying to say that their study is only relevant in pig but not in human or other organisms? Because that it what it sounds like. I’m not sure I would agree that this is a reasonable conclusion. The sentence should be clarified.
10) Line 211: “[…] which exerted no effect on cell viability.” How was this tested? Were control cells treated with 0.1% DMSO?
11) Line 215: The manufacturer and exact name or product number of the MTT assay kit should be provided.
12) Line 237: I could not access the repository PRJNA578240. As a reviewer I should have been given a reviewer token to access and review an unpublished dataset, but it seems this dataset does not exist at all.
13) Line 256: There are several ECL chemiluminescence kits and it is not clear which one was used. Product numbers should be provided.
Minor things, typos etc.
- In many of the figures, the significance of test results is encoded by asterisks. Doing so is an anachronistic relic from a time when figure typesetting was difficult. Today, exact P values should be provided wherever possible. There is nothing preventing one from stating exact P values above arcs connecting the test groups, which should be done as the higher, more rigorous scientific standard.
- Figure 2d x-axis label should read “Enrichment factor” or better “Fold enrichment” instead of “Rich factor” which is not a thing.
- Figure 5 y-axis label should read “Fold change over control” or something similar instead of “Folds of control”.
- Line 170: The term “evolution” means a concept in biology that is not meant here. This should be rephrased as “development” or “onset” or similar.
- Line 180: Under a “sequencing analysis” one would understand an analysis, in which the sequence is being analyzed in some way. Instead, I believe the authors refer to the RNA-seq analysis, which focuses much rather on RNA quantities than their exact sequences. This should be rephrased.
- The English language used in the text is not great, but usually understandable. I found two sentences that I did not understand at all and that should be fixed: Lines 32-34, “According to the dose …” and lines 190-191, “Caco-2 cells in answer …”
Best wishes!
Author Response
Response to Reviewer 1 Comments
Major comments:
Point 1: The GO and KEGG enrichment analyses are not described in sufficient detail. What methods were used to perform these analyses? What test sets were used? It appears from the results that the authors analyzed up-regulated and down-regulated DEG sets differently. I am not sure that is appropriate. In a pathway there are typically some genes that inhibit others, meaning that, for example, down-regulation of an inhibitor leads to up-regulation of its targets. In this case, down-regulation of the inhibitor would be important for pathway function. The authors should therefore repeat the analysis this way. I’m not saying the analysis as done is wrong, but also doing it the other way could add important details.
Response 1: Sorry for confusion, and we have modified the ambiguous text to be more clear. ClusterProfiler is a R package applied to perform GO function and KEGG pathway enrichment analyses on DEGs. The terms were considered to be significantly enriched if q-value ≤ 0.05 (line 268).
Genes with a q-value ≤0.05 and |fold change| ≥2 were defined as differentially expressed. The test sets were the same. But the down-regulated DEG enrichment was not obvious in KEGG pathway analysis.
Minor comments:
Point 2: Figure 1A has a lot of problems: Why is Figure 1A separated from the remaining panels? Why does the viability scale in Figure 1A go to 150%? There cannot be a viability >100%, so the axis should stop at 100%. The x-axis is linear, but the data is exponential. Therefore, the axis should be changed to exponential. I think it would make more intuitive sense to plot time on the x-axis and encode DON concentration in different symbols/colors, even though there are more concentrations than time points.
Response 2: As your comment, Figure 1A was combined with the rest of the panels and redrawn according to your suggestion (see Figure 1a).
Point 3: Statistical significance in the cell viability assay should not be calculated as separate tests per time point and concentration. Instead, these should be factors in a single, multivariate testing procedure (two-way ANCOVA).
Response 3: We agreed with your comments. Data of cell viability were analyzed by a two-way ANOVA using the Holm–Sidak method. We have revised the manuscript (see Figure 1a).
Point 4: Line 82-83: “These results suggest that DON-induced inflammation was associated with the MAPK signaling pathway.” No, they do not. At least not more than several other enriched GO terms. TNF signaling is much more strongly enriched than MAPK signaling. At this point, without the results presented in Figure 5 below, there is no reason to say that MAPK signaling is more enriched than others. Besides, MAPK signaling is one of several ways of how TNF signaling can be resolved in a cellular response, others being NF-kappa B signaling and Jak-STAT signaling, which are also enriched at similar levels as MAPK signaling.
Similarly, the sentence in lines 167-169 needs to be rephrased as it wrongly emphasizes MAPK signaling as the major outcome from the KEGG analysis, which is incorrect.
Response 4: Thank you for pointing out our mistakes. We have revised the manuscript according to your suggestion (line 97 and line 187).
Point 5: The legend of Figure 2c says this shows a “GO terms of biological function analysis.” This needs to be elaborated. What contrast is being used? Does it include all expressed genes or only DEGs? The term “biological function” reminds me of “molecular function” and “biological processes”, both of which are distinct GO subcategories, but all three of the subcategories are apparently tested. This should be phrased more cautiously.
Response 5: Thank you for pointing out our mistakes. We have revised legend of Figure 2c to “GO analysis classified the DEGs into 3 groups: molecular function, biological process, and cellular component” (see the legend of Figure 2c).
Point 6: Error bars should be provided for Figure 2e. Both qPCR and RNA-seq have been done in replicates, so error estimates for both techniques exist and should be provided.
Response 6: We have revised the manuscript according to your suggestion (see Figure 2e).
Point 7: Lines 102-101: The data underlying this sentence should be provided.
Response 7:
|
Category |
Term |
Count |
% |
PValue |
Genes |
List Total |
Pop Hits |
Pop Total |
Fold Enrichment |
Bonferroni |
Benjamini |
FDR |
|
KEGG_PATHWAY |
ssc04668:TNF signaling pathway |
8 |
38.095 |
2.93E-09 |
VCAM1, CSF2, FOS, CCL20, JUN, CXCL2, IL15, JUNB |
18 |
108 |
7001 |
28.8107 |
1.84E-07 |
1.84E-07 |
2.97E-06 |
|
KEGG_PATHWAY |
ssc05132:Salmonella infection |
6 |
28.571 |
1.02E-06 |
CSF2, FOS, JUN, CXCL2, CCL4, IL1A |
18 |
81 |
7001 |
28.8107 |
6.40E-05 |
3.20E-05 |
0.00102853 |
|
KEGG_PATHWAY |
ssc05323:Rheumatoid arthritis |
6 |
28.571 |
1.37E-06 |
CSF2, FOS, CCL20, JUN, IL15, IL1A |
18 |
86 |
7001 |
27.1357 |
8.64E-05 |
2.88E-05 |
0.00138801 |
|
KEGG_PATHWAY |
ssc05166:HTLV-I infection |
7 |
33.333 |
1.66E-05 |
EGR1, VCAM1, CSF2, FOS, ATF3, JUN, IL15 |
18 |
248 |
7001 |
10.9783 |
0.0010435 |
2.61E-04 |
0.01677871 |
|
KEGG_PATHWAY |
ssc04060:Cytokine-cytokine receptor interaction |
5 |
23.81 |
0.0011698 |
CSF2, CCL20, IL15, CCL4, IL1A |
18 |
201 |
7001 |
9.67523 |
0.0710863 |
0.01464 |
1.17816653 |
|
KEGG_PATHWAY |
ssc04010:MAPK signaling pathway |
4 |
19.048 |
0.0198016 |
FOS, DUSP1, JUN, IL1A |
18 |
244 |
7001 |
6.37614 |
0.7163511 |
0.118387 |
18.332483 |
|
KEGG_PATHWAY |
ssc04380:Osteoclast differentiation |
4 |
19.048 |
0.0037504 |
FOS, JUN, IL1A, JUNB |
18 |
133 |
7001 |
11.6976 |
0.2107859 |
0.038685 |
3.73313137 |
|
KEGG_PATHWAY |
ssc05140:Leishmaniasis |
3 |
14.286 |
0.0096344 |
FOS, JUN, IL1A |
18 |
62 |
7001 |
18.8199 |
0.4566017 |
0.083442 |
9.33755536 |
|
KEGG_PATHWAY |
ssc04610:Complement and coagulation cascades |
3 |
14.286 |
0.0128386 |
THBD, F3, SERPINE1 |
18 |
72 |
7001 |
16.206 |
0.5569475 |
0.096752 |
12.2641197 |
|
KEGG_PATHWAY |
ssc05133:Pertussis |
3 |
14.286 |
0.0131815 |
FOS, JUN, IL1A |
18 |
73 |
7001 |
15.984 |
0.5665391 |
0.088701 |
12.5722072 |
|
KEGG_PATHWAY |
ssc04620:Toll-like receptor signaling pathway |
3 |
14.286 |
0.024342 |
FOS, JUN, CCL4 |
18 |
101 |
7001 |
11.5528 |
0.7882857 |
0.131631 |
22.0829377 |
|
KEGG_PATHWAY |
ssc04660:T cell receptor signaling pathway |
3 |
14.286 |
0.0252487 |
CSF2, FOS, JUN |
18 |
103 |
7001 |
11.3285 |
0.8003313 |
0.125636 |
22.8130712 |
|
KEGG_PATHWAY |
ssc05142:Chagas disease (American trypanosomiasis) |
3 |
14.286 |
0.0271037 |
FOS, JUN, SERPINE1 |
18 |
107 |
7001 |
10.905 |
0.8229099 |
0.124676 |
24.287508 |
Point 8: Line 149: Source 7 references a review paper, not a study. If the authors intend to refer to a certain study, this original study should be cited instead of the review paper. Citing the review paper is only appropriate if it summarizes the results from several studies. In that case, the sentence needs to be rephrased to reflect this meaning.
Response 8: We have replaced the references 7 (line166).
Point 9: Line 192: “The discrepancy is possibly because of the cells origin.” Are the authors trying to say that their study is only relevant in pig but not in human or other organisms? Because that it what it sounds like. I’m not sure I would agree that this is a reasonable conclusion. The sentence should be clarified.
Response 9: We have revised the manuscript. This discrepancy may be due to the maturation status of the cells: differentiated mature Caco-2 cells and IPEC-J2 vs. undifferentiated Int407 cells (line 217).
Point 10: Line 211: “[…] which exerted no effect on cell viability.” How was this tested? Were control cells treated with 0.1% DMSO?
Response 10: We tested the cell viability with MTT. Besides, in our pre-experiment vehicle control with DMSO did not induce inflammation in IPEC-J2 Cells. The control was vehicle control in the study.
Point 11: Line 215: The manufacturer and exact name or product number of the MTT assay kit should be provided.
Response 11: We have revised the manuscript according to your suggestion (line 241).
Point 12: Line 237: I could not access the repository PRJNA578240. As a reviewer I should have been given a reviewer token to access and review an unpublished dataset, but it seems this dataset does not exist at all.
Response 12: The dataset really exists. We requested to update the release date for the BioProject below to 2020-11-28. SRA records will be accessible after the indicated release date.
Point 13: Line 256: There are several ECL chemiluminescence kits and it is not clear which one was used. Product numbers should be provided.
Response 13: We have provided the product number and revised the manuscript (line 285).
Minor things, typos etc.
Point 14: In many of the figures, the significance of test results is encoded by asterisks. Doing so is an anachronistic relic from a time when figure typesetting was difficult. Today, exact should be provided wherever possible. There is nothing preventing one from stating exact P values above arcs connecting the test groups, which should be done as the higher, more rigorous scientific standard.
Response 14: Thanks for your comments. But in our manuscript Figure 5 and Figure 6 were analyzed with different control. Provided exact P values would be a bit confusing. I did not find the suitable reference figure for proved P values. If you provide one, we are happy to correct the figure.
- Point 15: Figure 2d x-axis label should read “Enrichment factor” or better “Fold enrichment” instead of “Rich factor” which is not a thing.
Response 15: We have replaced Figure 2d x-axis label “Rich factor” by the word “Enrichment factor” (see Figure 2d).
Point 16: Figure 5 y-axis label should read “Fold change over control” or something similar instead of “Folds of control”.
Response 16: We have replaced y-axis label “Folds of control” by the word “Fold change over control” (see Figure 5).
Point 17: Line 170: The term “evolution” means a concept in biology that is not meant here. This should be rephrased as “development” or “onset” or similar.
Response 17: We have replaced the word “evolution” by the word “development” (line 190).
Point 18: Line 180: Under a “sequencing analysis” one would understand an analysis, in which the sequence is being analyzed in some way. Instead, I believe the authors refer to the RNA-seq analysis, which focuses much rather on RNA quantities than their exact sequences. This should be rephrased.
Response 18: We appreciate the review’s attention to detail, and we have corrected the description as suggested (line 199).
Point 19: The English language used in the text is not great, but usually understandable. I found two sentences that I did not understand at all and that should be fixed: Lines 32-34, “According to the dose …” and lines 190-191, “Caco-2 cells in answer …”
Response 19: We have undergone extensive English editing.
Special thanks to you for your good comments.
Reviewer 2 Report
The paper showed deep insight into the molecular mechanism of inflammatory pathways as effect of DON in a porcine intestinal cell in vitro system.
The paper is well written and the methodology is correct.
There are only some explanation is missing, which are requires for correct evaluation of data.
- 59-65 Please mention here that the highest DON concentration did not use for expression studies.
- 73 - Please indicate the cause to use only 2 mg/ml DON concentration for functional expression analysis, which means that the data based on only that concentration, but the dose related changes remains unknown
- 110 - Please explain the cause that only one dose (2 mg/ml) DON dose was selected for the investigation of the expression of inflammatory factors
Additional comment:
- 26-27. Accumulation of DON in edible animal tissues is negligible (except liver), therefore the statement about its dangerous effect in the food chain through animal-origin foods is not correct.
Author Response
Point1、59-65 Please mention here that the highest DON concentration did not use for expression studies.
Response 1: We observed that most cells were treatment with 2 μg/mL and below, and 2.0 μg/mL DON have induced obviously inflammation in IPEC-J2 cells. So the highest DON concentration 4.0 μg/mL did not use for expression studies.
Chen, Ying, et al. Nontoxic concentrations of OTA aggravate DON-induced intestinal barrier dysfunction in IPEC-J2 cells via activation of NF-κB signaling pathway.[J]. Toxicology letters, 2019. 4mM(1.2 μg/mL)DON
Kang, R, Deoxynivalenol induced apoptosis and inflammation of IPEC-J2 cells by promoting ROS production. Environ Pollut 2019 0.5 1 2 μg/ mL DON
Simona A,et al. The Food Contaminants Nivalenol and Deoxynivalenol Induce Inflammation in Intestinal Epithelial Cells by Regulating Reactive Oxygen Species Release[J]. Nutrients, 2017 0.5–5 μM DON
Point2、73 - Please indicate the cause to use only 2 mg/ml DON concentration for functional expression analysis, which means that the data based on only that concentration, but the dose related changes remains unknown.
Response 2: We agreed with your comments about the one concentration. We did not consider the study well when designing the experiment, and we will pay attention to improvement in future research. It can be seen from the results (Fig.1) that DON induce inflammation in IPEC-J2 cells in a concentration- dependent manner.
Point3、110 - Please explain the cause that only one dose (2 mg/ml) DON dose was selected for the investigation of the expression of inflammatory factors.
Response 3: It can be seen from the results (Fig.1) that DON induce inflammation in IPEC-J2 cells in a concentration- dependent manner. So we selected higter concentration 2.0 to investigate of the expression of inflammatory factors. In our pre-experiment DON 1.0 also increased the phosphorylation of p38, ERK1/2 and JNK and both p38 and ERK 1/2 contribute to DON-induced inflammation.
Additional comment:
Point4、26-27. Accumulation of DON in edible animal tissues is negligible (except liver), therefore the statement about its dangerous effect in the food chain through animal-origin foods is not correct.
Response 4: Thank you for pointing out our mistakes. We have deleted this sentence (line 27).
Special thanks to you for your good comments.